# Gene-based analyses of the maternal genome implicate maternal effect genes as risk factors for conotruncal heart defects

Anshuman Sewda[1¤], A. J. Agopian[1], Elizabeth Goldmuntz[2,3], Hakon Hakonarson[2,4], Bernice E. Morrow[5], Fadi Musfee[1], Deanne Taylor[2,6], Laura E. Mitchell[1]*, on behalf of the Pediatric Cardiac Genomics Consortium[¶]

**1** Department of Epidemiology, Human Genetics and Environmental Sciences, UTHealth School of Public Health, Houston, Texas, United States of America, **2** Department of Pediatrics, University of Pennsylvania Perelman School of Medicine, Philadelphia, Pennsylvania, United States of America, **3** Division of Cardiology, The Children's Hospital of Philadelphia, Philadelphia, Pennsylvania, United States of America, **4** Center for Applied Genomics, The Children's Hospital of Philadelphia, Philadelphia, Pennsylvania, United States of America, **5** Department of Genetics, Albert Einstein College of Medicine, Bronx, New York, United States of America, **6** Department of Biomedical and Health Informatics, The Children's Hospital of Philadelphia, Philadelphia, Pennsylvania, United States of America

¤ Current address: Institute of Health Management Research, IIHMR University, Jaipur, Rajasthan, India
¶ Names and affiliations for members of the Pediatric Cardiac Genomics Consortium are provided in Acknowledgments.
* Laura.E.Mitchell@uth.tmc.edu

**Data Availability Statement:** Data underlying the figures in this manuscript are provided in the Supporting Information as follows: Fig 1 (Table D

## Abstract

Congenital heart defects (CHDs) affect approximately 1% of newborns. Epidemiological studies have identified several genetically-mediated maternal phenotypes (e.g., pregestational diabetes, chronic hypertension) that are associated with the risk of CHDs in offspring. However, the role of the maternal genome in determining CHD risk has not been defined. We present findings from gene-level, genome-wide studies that link CHDs to maternal effect genes as well as to maternal genes related to hypertension and proteostasis. Maternal effect genes, which provide the mRNAs and proteins in the oocyte that guide early embryonic development before zygotic gene activation, have not previously been implicated in CHD risk. Our findings support a role for and suggest new pathways by which the maternal genome may contribute to the development of CHDs in offspring.

## Introduction

Congenital heart defects (CHDs) are the most common group of birth defects, with a prevalence of approximately 1% in live births [1]. CHDs are also the leading cause of birth defect-related mortality [2] and account for the largest percentage of birth defect-associated hospitalizations and healthcare costs [3]. As for many birth defects, the risk of CHDs is associated with several genetically-mediated, maternal phenotypes, including folate status, obesity, pregestational diabetes, chronic hypertension, and preeclampsia [4, 5]. These associations suggest that the maternal genotype may contribute to the risk of birth defects in offspring, independent of

of S1 File); S1 Fig (Table B of S1 File); S2 Fig (Table C of S1 File). The genotype data used in these studies are available at: Pediatric Cardiac Genomics Consortium: https://www.ncbi.nlm.nih.gov/projects/gap/cgi-bin/study.cgi?study_id=phs001194.v2.p2 CHOP pediatric controls: https://www.ncbi.nlm.nih.gov/projects/gap/cgi-bin/study.cgi?study_id=phs000490.v1.p1 CHOP CTD trios: https://www.ncbi.nlm.nih.gov/projects/gap/cgi-bin/study.cgi?study_id=phs000881.v1.p1

**Funding:** This work was supported by grants from the Eunice Kennedy Shriver National Institute of Child Health and Human Development (P01HD070454, AJA, EG, BM, LEM, FM, AS, DT; R21HD097347, AJA, LEM; R03HD098552, LEM) (https://www.nichd.nih.gov/), the National Heart, Lung, and Blood Institute (P50-HL74731) (https://www.nhlbi.nih.gov/), including the PCGC (U01-HL098188, U01HL131003, U01-HL098147, U01-HL098153, U01-HL098163, U01-HL098123, U01-HL098162) (https://benchtobassinet.com) and the Cardiovascular Development Consortium (U01-HL098166) (https://benchtobassinet.com/?page_id=1644), as well as the National Human Genome Research Institute (U54HG006504) (https://www.genome.gov), and the National Center for Research Resources (M01-RR-000240, RR024134, which is now the National Center for Advancing Translational Sciences (UL1TR000003) (https://ncats.nih.gov). The content is solely the responsibility of the authors and does not necessarily represent the official views of the funding sources. Array genotyping of the CHOP cohorts was funded by an Institutional Development Fund to The Center for Applied Genomics from The Children's Hospital of Philadelphia. The funders had no role in study design, data collection and analysis, decision to publish, or preparation of the manuscript.

**Competing interests:** The authors have declared that no competing interests exist.

the maternal alleles transmitted to the child. For example, maternal genes involved in folate transport and metabolism may influence the availability of folate to the embryo, which in turn influences the risk of folate-related birth defects.

While there has been some interest in assessing the relationship between birth defects and maternal genotypes (e.g., methylenetetrahydrofolate reductase or MTHFR genotypes) [6–10], studies of the maternal genotype have considered a relatively small number of maternal phenotypes and are limited by gaps in our understanding of the genetic contribution to these phenotypes. Further, studies focused on maternal phenotypes ignore maternal genes that might act through alternate mechanisms to influence the risk of birth defects. For example, studies in model systems indicate that mutations in maternal effect genes (MEGs), which provide the mRNAs and proteins in the oocyte that guide early embryonic development before activation of the embryonic genome, can result in birth defects in offspring [11–13]. While genome-wide association studies (GWAS) provide a comprehensive, agnostic approach for identifying disease associations, only a few GWAS have focused on the maternal genotype [14–17]. Consequently, there is much to be learned about the role of maternal genes in determining the risk of birth defects such as CHDs.

We have previously conducted a single nucleotide polymorphism (SNP)-based GWAS of maternal genetic effects for conotruncal heart defects (CTDs) [14], which affect the cardiac outflow tracts [18] and account for approximately one-third of all CHDs [19]. Although we identified several maternal SNPs with suggestive evidence of association ($p \leq 10^{-5}$) with CTDs, no association was genome-wide significant ($p < 5 \times 10^{-8}$). Compared to SNP-based GWAS, gene-based GWAS has the advantage of a less stringent threshold for statistical significance. Furthermore, gene-based analyses can include both common and rare variants [20] and, therefore, capture a greater proportion of the within gene variation than SNP-based analyses, which generally exclude variants with minor allele frequencies (MAFs) less than 5% [21]. Given these advantages, we have undertaken gene-based GWAS and meta-analyses using data from two large CTD datasets to identify maternal genes associated with the risk of CTDs in offspring.

## Materials and methods

### Study subjects

**The Children's Hospital of Philadelphia (CHOP).** Patients with CTDs and their parents were recruited through the Cardiac Center at CHOP (1992–2010), under a protocol approved by the Institutional Review Board for the Protection of Human Subjects at CHOP [14, 15]. Adult participants provided written consent for themselves and their participating minor children.

Patients with the following diagnoses were eligible to be a CTD case: tetralogy of Fallot, persistent truncus arteriosus, D-transposition of the great arteries, double outlet right ventricle, ventricular septal defects (conoventricular, posterior malalignment, and conoseptal hypoplasia types), aortic-pulmonary window, interrupted aortic arch, and isolated aortic arch anomalies. Medical records, imaging (e.g., echocardiography and cardiac magnetic resonance imaging), and operative reports were used to confirm cardiac diagnoses. Potential cases were tested for the 22q11.2 deletion syndrome using fluorescence *in situ* hybridization, multiplex ligation-dependent probe amplification, or both, and those with a deletion were excluded [22]. Potential cases were also excluded if they had a clinically diagnosed chromosome abnormality or single-gene mutation.

**Pediatric Cardiac Genomics Consortium (PCGC).** Patients with CTDs and their parents were recruited as part of the PCGC Congenital Heart Defect GEnetic NEtwork Study (2010–

2012) [23]. Recruitment took place at five main (including CHOP) and four satellite clinical sites. Informed consent was obtained under protocols approved by the Institutional Review Board for each study site. Adult participants provided written consent for themselves and their participating minor children.

Patients recruited through the PCGC included those with the same CTD diagnoses as listed above. Cardiac diagnoses were confirmed through review of medical records and electronic case reports, and potential CTD cases were excluded if they had a clinically diagnosed chromosomal or genetic disorder. Participants recruited at CHOP as part of the PCGC do not overlap with the CHOP participants described above.

## Genetic methods

Blood samples were collected from cases, and blood or saliva samples were collected from parents of cases. When blood collection was scheduled in conjunction with a surgical procedure, the sample was collected before any blood transfusion. DNA extraction was performed using standard techniques. Genome-wide microarray genotyping was performed at the CHOP Center for Applied Genomics [14]. The CHOP samples were genotyped using the Illumina HumanOmni-2.5, Illumina HumanHap550 (v2 or v3), or 610 BeadChip platforms, and the PCGC samples were genotyped on the Illumina HumanOmni-1 or 2.5 platforms.

**Imputation and Quality Control (QC) procedures.** Standard QC procedures were performed for each dataset using Plink version 1.07 [24] and have been previously described [25]. Before imputation, the genotype data were checked for strand and coding errors. Case-parent trios were removed if more than 1% of genotyped SNPs had Mendelian errors. Suspected duplicate samples were identified using pairwise identity-by-descent estimation, and samples with pi-hat greater than 0.6 were removed. Samples with genotyping rates less than 95% were also removed. In addition, SNPs with MAF less than 1%, genotyping rates less than 90%, and all non-autosomal variants were excluded.

Due to differences in microarray genotyping platforms, the CHOP and PCGC case-parent trios data were imputed separately. After the pre-imputation exclusions, the CHOP data from different platforms (HumanOmni-2.5, HumanHap550K v2, 550K v3, and 610K) were combined, and the SNPs present across all platforms (N = 283,977 SNPs) were used for imputation. Similarly, the PCGC data from different platforms (HumanOmni-1 and HumanOmni-2.5) were combined, and the SNPs present on both platforms (N = 624,419 SNPs) were used for imputation.

For each dataset, haplotypes were pre-phased using SHAPEIT2 v2.727 [26], and imputation was performed using Impute2 v2.3.0 [27] with pre-phased haplotype data from the 1000 Genomes Project (version: Phase-I integrated v3 variants set) as the reference population. A genotype was imputed, only if the posterior probability value exceeded 0.9, the default calling threshold for Impute2. After imputation, we excluded SNPs with poor imputation quality (Impute2 information metric score less than 0.8), or genotyping rates less than 90%. Samples with genotyping rates less than 95% were removed. Because we were interested in assessing both common and rare SNPs, the post-imputation QC procedures did not include restrictions based on MAFs.

## Statistical analysis

**Genome-wide gene-based analyses.** Maternal genetic effects were evaluated using a case-control approach in which mothers and fathers from the CTD trios were considered as cases and controls, respectively. Genes were defined by their transcription start-stop positions, including untranslated regions (hg19 reference assembly) plus 1kb upstream and downstream.

Analyses were conducted separately for the CHOP and PCGC datasets, using the sequence kernel association test for the combined effect of common and rare variants (SKAT-C) [28]. In this approach, separate scores were calculated for the common (MAF $\geq$ 5%) and rare (MAF < 5%) variants in each gene, and p-values were based on the weighted sum of these scores. We used the SKAT-C default parameters for weighting common and rare SNPs and evaluated all autosomal genes with at least one common and one rare variant in our data.

To control for population stratification bias, only the parents of non-Hispanic Caucasian CTD cases (based on self- or parental-report) were included in the analyses. As race/ethnicity was based on the report rather than genetic data, we adjusted for the first genotypic principal component. Genotypic principal components analyses were conducted using Golden Helix SVS version 8.1 (Golden Helix, Inc., Bozeman, Montana, USA; www.goldenhelix.com), using the default parameter settings (additive genetic model, MAF-based allele classification, and each marker data normalized by its theoretical standard deviation under Hardy Weinberg Equilibrium). A meta-analysis of the gene-based results from the CHOP and PCGC datasets was performed using Fisher's combination of probability method [29]. For each analysis, the genomic inflation factor ($\lambda$) was calculated, and quantile-quantile (Q-Q) plots were constructed to check for deviation of the observed distribution of the test statistic from the expected null distribution.

An association was considered genome-wide significant if the meta-analysis p-value was less than the Bonferroni-corrected p-value, based on the number of genes evaluated. Genes with meta-analysis $p < 10^{-3}$ were considered to have suggestive evidence of association. For genes with at least suggestive evidence of association in the meta-analysis, we considered those for which the meta-analysis p-value was lower than the p-values in the contributing datasets (i.e., the evidence for an association was stronger in the combined data than in either of the individual datasets) as candidate maternal CTD-related genes. When several contiguous genes met these criteria, which may reflect linkage disequilibrium between variants in genes that are in close proximity rather than independent association signals, we reviewed gene functions [30] to identify the most likely candidate gene in the region.

**Gene-set enrichment analyses.** Enrichment analyses using MetaCore™ (Thomson Reuters, Life Science Research; https://portal.genego.com/metacore)), were performed for genes with meta-analysis $p < 0.01$ to identify enriched gene ontology (GO) processes, diseases (represented by biological markers), pathway maps, and pathway processes. For these analyses, a false-discovery rate (FDR)-corrected $p < 0.05$ was considered statistically significant.

**Post hoc analyses of maternal effect genes (MEGs).** The most significant association in our meta-analysis was with a gene that has been suggested to be a MEG [31]. Given this finding, we elected to conduct an *a posteriori*, MEG gene-set analysis. For this analysis, we considered a gene to be an established MEG if it was included in at least one of two comprehensive reviews of the MEG literature (Table A of S1 File) [32, 33]. Fisher's exact test was used to compare the proportion of established MEGs among all genes with meta-analysis p-values below and above a specified p-value cut-point (i.e., $p < 0.05$ versus $p \geq 0.05$). A Fisher's exact $p < 0.05$ was considered statistically significant. In addition, we cross-referenced the list of established MEGs with our list of candidate maternal CTD-related genes.

## Results

In both the CHOP and PCGC datasets, the most common diagnosis in the offspring was the tetralogy of Fallot (Table 1). After QC exclusions, the CHOP dataset included 423 mothers and 380 fathers, and the PCGC dataset included 216 mothers and 219 fathers.

**Table 1. Summary of the conotruncal heart defect phenotypes in the offspring of study subjects.**

| Conotruncal Heart Defect Phenotype | CHOP | | PCGC | |
|---|---|---|---|---|
| | N = 483 | % | N = 244 | % |
| Tetralogy of Fallot | 196 | 40.6 | 73 | 29.9 |
| D-transposition of the great arteries | 95 | 19.7 | 52 | 21.3 |
| Ventricular septal defects | 90 | 18.6 | 34 | 13.9 |
| Double outlet right ventricle | 53 | 11.0 | 37 | 15.2 |
| Isolated aortic arch anomalies | 22 | 4.6 | 7 | 2.9 |
| Truncus arteriosus | 15 | 3.1 | 7 | 2.9 |
| Interrupted aortic arch | 6 | 1.2 | 6 | 2.4 |
| Other | 6 | 1.2 | 28 | 11.5 |

Abbreviations: CHOP, The Children's Hospital of Philadelphia; PCGC, The Pediatric Cardiac Genomics Consortium.

The number of variants and genes included in the CHOP and PCGC datasets are summarized in Table 2. The Q-Q plots (S1 and S2 Figs) and genomic inflation factors (Table 2) for the analyses of the individual datasets provided little evidence for systematic bias (Tables C and D of S1 File). No genome-wide significant associations ($p \leq 2.3 \times 10^{-6}$) were detected in either dataset.

Fisher's method was used to conduct a meta-analysis of the SKAT-C p-values from the 20,962 genes (Table D of S1 File) that were analyzed in both the CHOP and PCGC datasets. The genomic inflation factor ($\lambda = 1.07$) and Q-Q plot provided little evidence of a systematic deviation from the expected distribution (Fig 1). Although no gene achieved genome-wide significance in the meta-analysis (Bonferroni-corrected $p < 2.4 \times 10^{-6}$), the meta-analysis p-value for the germ cell-specific gene, *GGN*, was of borderline significance ($p = 7.1 \times 10^{-6}$). The meta-analysis also provided suggestive evidence of association for an additional 30 genes (Table 3).

Of the 31 genes with suggestive evidence for association, ten had meta-analysis p-values lower than the p-values in either individual dataset. These ten genes included one pseudogene (*TBC1D29P*), one RNA gene (*LOC101928565*), and eight protein-coding genes. The eight protein-coding genes include two contiguous genes located at 3q22.1 (*H1FOO* and *PLXND1*); *SLAIN2* at 4p11; and five genes located in an approximately 100,000 base-pair region of 19q13.2 (*YIF1B*, *CATSPERG*, *PSMD8*, *GGN*, and *SPRED3*) (Table 4). Based on their known functions (Table 4), the eight protein-coding genes do not appear to be strong candidates for maternal genes that act via a maternal phenotype (e.g., obesity and diabetes). However, the 3q22.1 region includes a known MEG, *H1FOO* (meta-p = $7.9 \times 10^{-4}$), and the 19q13.2 region

**Table 2. Summary of the genetic data used in the analyses of the CHOP and PCGC datasets.**

| | Dataset (# mothers/# fathers) | |
|---|---|---|
| | CHOP (423/380) | PCGC (216/219) |
| Total variants | 5,605,644 | 6,815,834 |
| Rare variants[a] | 3,500,915 | 4,574,369 |
| Number of genes | 21,187 | 22,002 |
| Genomic inflation factor ($\lambda$) | 1.06 | 1.05 |

Abbreviations: CHOP, The Children's Hospital of Philadelphia; PCGC, The Pediatric Cardiac Genomics Consortium.

[a] Variants with minor allele frequency < 0.05.

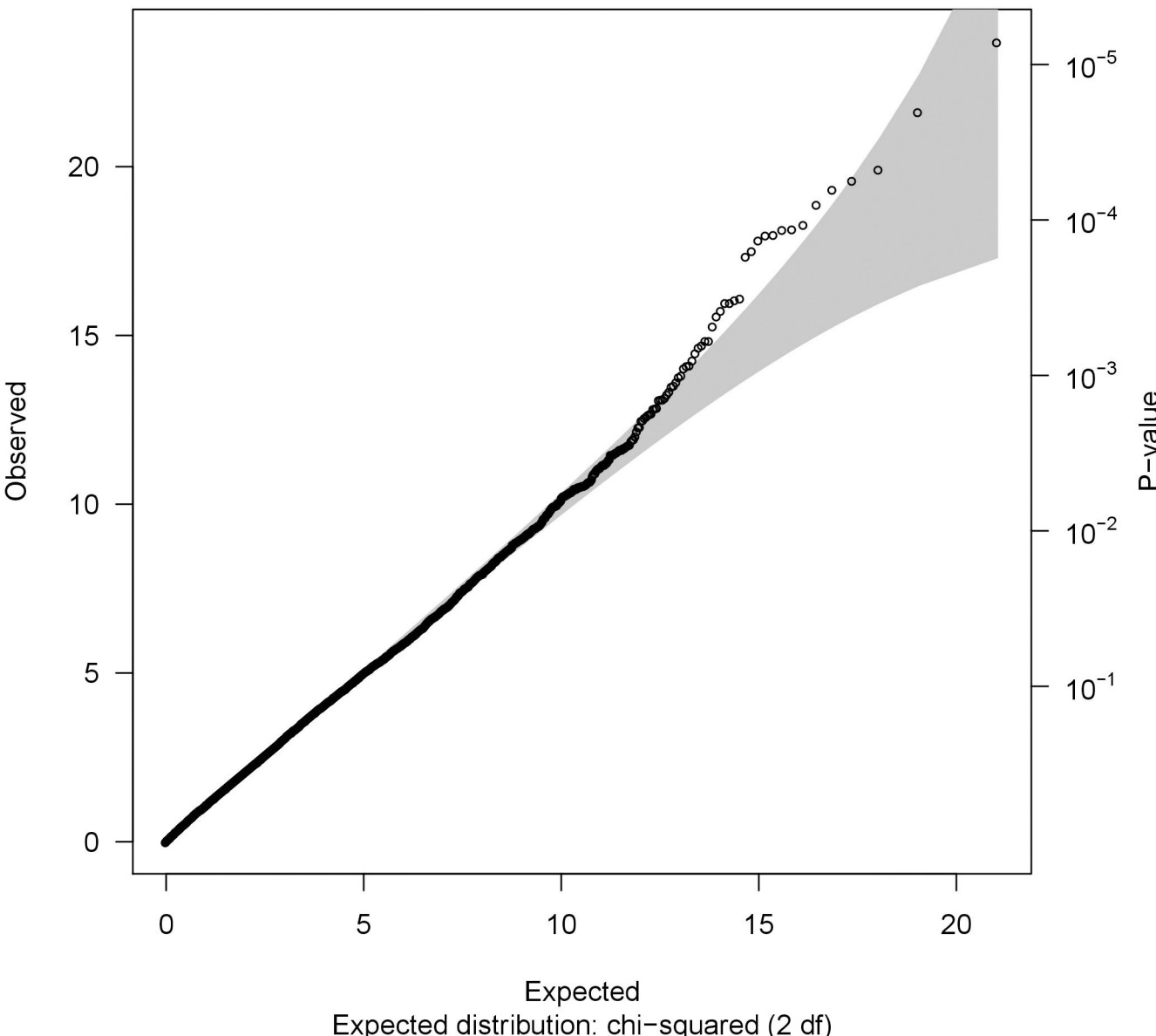

**Fig 1. A quantile-quantile plot.** A quantile-quantile plot of meta-analysis p-values obtained by combining SKAT-C test p-values from genome-wide analyses of the CHOP and PCGC datasets.

includes a gene that has been proposed to be a MEG, *GGN* (meta-p = 7.1 $\times 10^{-6}$). Consequently, we propose *H1FOO* and *GGN*, as well as *SLAIN2* (the single associated gene in the 4p11 region) as the top candidate maternal CTD-related genes identified by our meta-analysis.

## Gene-set enrichment analyses

In enrichment analyses of genes with meta-analysis p < 0.01 (N = 204 genes), no pathway map or pathway process was significant. However, there was evidence of enrichment (FDR p < 0.05) for 17 GO processes (Table E of S1 File), including several related to transmembrane transport in general, and calcium ion transport in particular (e.g., GO:1903169, regulation of calcium ion transmembrane transport, p = 2.9 $\times 10^{-2}$). There was also evidence for enrichment

**Table 3. Maternal genes with suggestive evidence of association ($p < 10^{-3}$) with conotruncal heart defects in the meta-analysis.**

| Gene | CHR | CHOP | | PCGC | | Meta-analysis[a] |
|---|---|---|---|---|---|---|
| | | # of variants | p-value | # of variants | p-value | p-value |
| *GGN* | 19 | 17 | $6.30 \times 10^{-4}$ | 18 | $7.23 \times 10^{-4}$ | $7.10 \times 10^{-6}$ |
| *SPRED3* | 19 | 46 | $5.11 \times 10^{-3}$ | 44 | $2.71 \times 10^{-4}$ | $2.01 \times 10^{-5}$ |
| *VARS2* | 6 | 88 | $7.29 \times 10^{-6}$ | 84 | $4.76 \times 10^{-1}$ | $4.71 \times 10^{-5}$ |
| *FER1L6-AS1* | 8 | 78 | $5.66 \times 10^{-6}$ | 153 | $7.30 \times 10^{-1}$ | $5.54 \times 10^{-5}$ |
| *LOC101927269* | 7 | 18 | $2.80 \times 10^{-1}$ | 21 | $1.71 \times 10^{-5}$ | $6.34 \times 10^{-5}$ |
| *LOC151475* | 2 | 25 | $1.49 \times 10^{-5}$ | 31 | $4.08 \times 10^{-1}$ | $7.91 \times 10^{-5}$ |
| *SUMO1* | 2 | 84 | $7.48 \times 10^{-1}$ | 74 | $1.12 \times 10^{-5}$ | $1.06 \times 10^{-4}$ |
| *PSMD8* | 19 | 40 | $8.51 \times 10^{-4}$ | 39 | $1.06 \times 10^{-2}$ | $1.14 \times 10^{-4}$ |
| *SPINT4* | 20 | 43 | $4.38 \times 10^{-1}$ | 41 | $2.08 \times 10^{-5}$ | $1.15 \times 10^{-4}$ |
| *SLAIN2* | 4 | 292 | $1.11 \times 10^{-2}$ | 262 | $8.89 \times 10^{-4}$ | $1.23 \times 10^{-4}$ |
| *YIF1B* | 19 | 52 | $3.20 \times 10^{-2}$ | 58 | $3.12 \times 10^{-4}$ | $1.25 \times 10^{-4}$ |
| *CATSPERG* | 19 | 162 | $4.72 \times 10^{-3}$ | 186 | $2.29 \times 10^{-3}$ | $1.34 \times 10^{-4}$ |
| *FER1L6* | 8 | 785 | $4.51 \times 10^{-5}$ | 897 | $2.84 \times 10^{-1}$ | $1.57 \times 10^{-4}$ |
| *LOC101928565* | 1 | 113 | $4.45 \times 10^{-2}$ | 163 | $3.14 \times 10^{-4}$ | $1.70 \times 10^{-4}$ |
| *SFTA2* | 6 | 33 | $1.01 \times 10^{-4}$ | 31 | $2.73 \times 10^{-1}$ | $3.18 \times 10^{-4}$ |
| *PTPRF* | 1 | 342 | $3.56 \times 10^{-1}$ | 317 | $7.95 \times 10^{-5}$ | $3.25 \times 10^{-4}$ |
| *PLXND1* | 3 | 224 | $1.26 \times 10^{-3}$ | 208 | $2.35 \times 10^{-2}$ | $3.38 \times 10^{-4}$ |
| *TBC1D29* | 17 | 19 | $2.57 \times 10^{-2}$ | 12 | $1.16 \times 10^{-3}$ | $3.40 \times 10^{-4}$ |
| *KDM4A* | 1 | 161 | $3.21 \times 10^{-1}$ | 120 | $1.05 \times 10^{-4}$ | $3.81 \times 10^{-4}$ |
| *TNK2* | 3 | 100 | $9.79 \times 10^{-2}$ | 99 | $3.77 \times 10^{-4}$ | $4.14 \times 10^{-4}$ |
| *ZSWIM3* | 20 | 110 | $2.52 \times 10^{-1}$ | 98 | $1.72 \times 10^{-4}$ | $4.80 \times 10^{-4}$ |
| *LOC100505978* | 12 | 8 | $1.32 \times 10^{-1}$ | 10 | $4.15 \times 10^{-4}$ | $5.92 \times 10^{-4}$ |
| *MYDGF* | 19 | 71 | $1.21 \times 10^{-4}$ | 80 | $4.52 \times 10^{-1}$ | $5.92 \times 10^{-4}$ |
| *FTH1* | 11 | 14 | $1.93 \times 10^{-4}$ | 8 | $3.05 \times 10^{-1}$ | $6.33 \times 10^{-4}$ |
| *WFDC13* | 20 | 27 | $6.72 \times 10^{-1}$ | 26 | $9.08 \times 10^{-5}$ | $6.53 \times 10^{-4}$ |
| *WFDC3* | 20 | 89 | $5.44 \times 10^{-1}$ | 77 | $1.24 \times 10^{-4}$ | $7.13 \times 10^{-4}$ |
| *H1FOO* | 3 | 50 | $2.16 \times 10^{-2}$ | 46 | $3.50 \times 10^{-3}$ | $7.92 \times 10^{-4}$ |
| *TBX20* | 7 | 102 | $2.66 \times 10^{-4}$ | 124 | $3.10 \times 10^{-1}$ | $8.58 \times 10^{-4}$ |
| *ZNF622* | 5 | 40 | $1.96 \times 10^{-4}$ | 45 | $4.24 \times 10^{-1}$ | $8.62 \times 10^{-4}$ |
| *HPS3* | 3 | 193 | $4.54 \times 10^{-4}$ | 193 | $1.90 \times 10^{-1}$ | $8.91 \times 10^{-4}$ |
| *STARD7-AS1* | 2 | 37 | $3.53 \times 10^{-1}$ | 35 | $2.74 \times 10^{-4}$ | $9.92 \times 10^{-4}$ |

Abbreviations: CHR, chromosome; CHOP, The Children's Hospital of Philadelphia; PCGC, The Pediatric Cardiac Genomics Consortium.

[a]The meta-analysis included 20,992 genes.

of genes for biological markers associated with 24 disease processes including, diseases of proteostasis (e.g., proteostasis deficiencies, $p = 8.2 \times 10^{-3}$) and hypertension (FDR $p = 3.0 \times 10^{-2}$) (Table F of S1 File).

## Post hoc analyses of MEGs

Given that the most significant association in our meta-analysis was with *GGN* ($p = 7.1 \times 10^{-6}$), a gene that has been suggested to be a MEG [31], we elected to conduct an *a posteriori*, MEG gene-set analysis. Based on two comprehensive reviews of the MEG literature [32, 33], we identified a list of 60 MEGs (Table A of S1 File). In our meta-analysis six of the genes on this list had $p < 0.05$ (*HF1OO*, $p = 0.0008$; *KMT2D*, $p = 0.015$; *TP73*, $p = 0.026$; *BNC1*, $p = 0.034$; *ZAR1*, $p = 0.036$; and *RNF2*, $p = 0.0497$). Although *GGN* also had a meta-analysis

**Table 4. Maternal protein-coding genes with meta-analysis p-values suggestive of association (p $<$ 10$^{-3}$) and lower than the p-values from the analysis of individual datasets.**

| Chr. | Gene | Position[a] | Description[b] | Maternal Effect Gene | CHOP p-value | PCGC p-value | Meta-analysis p-value |
|---|---|---|---|---|---|---|---|
| 19q13.2 | YIF1B | 38,793,200–38,807,606 | Membrane trafficking protein | | $3.20 \times 10^{-2}$ | $3.12 \times 10^{-2}$ | $1.25 \times 10^{-4}$ |
| | CATSPERG | 38,825,443–38,862,589 | Sub-unit of the sperm calcium channel, CATSPER | | $4.72 \times 10^{-3}$ | $2.29 \times 10^{-3}$ | $1.34 \times 10^{-4}$ |
| | PSMD8 | 38,864,190–38,875,464 | Involved in ATP-dependent degradation of ubiquinated proteins | | $8.51 \times 10^{-4}$ | $1.06 \times 10^{-2}$ | $1.14 \times 10^{-4}$ |
| | GGN | 38,873,992–38,879,668 | Germ cell specific gene | Suggested | $6.30 \times 10^{-4}$ | $7.23 \times 10^{-4}$ | $7.10 \times 10^{-6}$ |
| | SPRED3 | 38,879,840–38,891,523 | Negative regulation of MAP kinase signaling | | $5.11 \times 10^{-3}$ | $2.71 \times 10^{-4}$ | $2.01 \times 10^{-5}$ |
| 3q22.1 | H1FOO | 129,261,057–129,271,310 | Oocyte specific member of the H1 histone family | Established | $2.16 \times 10^{-2}$ | $3.40 \times 10^{-3}$ | $7.92 \times 10^{-4}$ |
| | PLXND1 | 129,273,056–129,326,582 | Cell surface receptor for semaphorins | | $1.26 \times 10^{-3}$ | $2.35 \times 10^{-2}$ | $3.38 \times 10^{-4}$ |
| 4p11 | SLAIN2 | 48,342,613–48,429,215 | Promotes cytoplasmic microtubule nucleation and elongation | | $1.11 \times 10^{-2}$ | $8.89 \times 10^{-4}$ | $1.23 \times 10^{-4}$ |

Abbreviations: CHOP, The Children's Hospital of Philadelphia; PCGC, The Pediatric Cardiac Genomics Consortium.

[a] Gene transcription start/stop positions (hg19) plus 1 kb upstream and downstream.

[b] Gene descriptions obtained from GeneCards: https://www.genecards.org/.

p $<$ 0.05, this gene was not included in either of the review articles and was therefore omitted from these analyses. The identification of six MEGs with meta-analysis p $<$ 0.05 represents a 2.3-fold enrichment, which is of borderline significance (Fisher's exact p = 0.057) based on the standard p-value cut-off for a single statistical test (i.e. p $<$ 0.05).

## Discussion

Our genome-wide, gene-based analyses of common and rare variants provide suggestive evidence that maternal genes are associated with the risk of CTDs in their offspring. Based on the analyses of individual genes, we identified three candidate CTD-related maternal genes, *H1FOO*, *GGN*, and *SLAIN2*, and propose that these genes are most likely to influence CTD-risk via effects on early embryonic development.

*H1FOO* (meta-p = $7.1 \times 10^{-6}$), a known MEG [33], is an oocyte-specific member of the linker histone H1 family [34]. Genes in this family are involved in the determination of higher-order chromatin structure and gene transcription. Knockdown studies of *H1foo* in mouse one-cell embryos indicate that maternal *H1foo* influences the progression of DNA replication by reducing the deposition of H3 in the perinuclear region of the male pronucleus, and significantly delays the timing of cleavage into a two-cell embryo [35].

The suspected MEG, *GGN*, is thought to be involved in DNA repair and is characterized as a germ cell-specific gene. *GGN* is expressed at high levels in the adult testis [36], and at lower levels in the adult ovary and somatic tissues, as well as in Metaphase-II (MII) oocytes and early embryos [36, 37]. Evidence that *GGN* may function as a MEG is based on the timing of the loss of viability in mouse *Ggn*$^{-/-}$ embryos. Specifically, *Ggn*$^{-/-}$ embryos are present in expected numbers at the two-cell stage but are rarely observed at the morula stage and absent by embryonic day 7.5, consistent with the loss of viability following the depletion of maternal *Ggn* mRNA stores [31].

Although *SLAIN2* has not previously been implicated as a MEG, *SLAIN2* mRNA is abundant in both MII oocytes and one-cell embryos and declines thereafter [37]. Further, *SLAIN2* is involved in microtubule dynamics and organization [38], which are essential for several post-fertilization processes, including meiotic spindle assembly, separation of the parental genomes, and pronuclei migration [39–42]. Hence, both the expression pattern and known functions of *SLAIN2* are compatible with a potential role as a MEG.

Additional evidence that MEGs may be associated with CTDs in offspring is provided by the observed 2.3-fold enrichment of established MEGs among genes with p < 0.05 in our meta-analysis. The established MEGs with meta-analysis p < 0.05 include: *H1FOO* (discussed above); the transcriptional regulators, *BNC1* and *KMT2D*; *RNF2*, which is involved in chromatin remodeling; and, *TP73* and *ZAR1*, which are involved in cell cycle regulation [32]. Although MEGs have not previously been implicated as potential maternal risk factors for CTDs, studies in model systems demonstrate that mutations in MEGs can have a range of consequences for offspring, including embryonic lethality, developmental delay, and congenital malformations [11–13]. Similarly, women carrying a MEG mutation (e.g., *NLRP5*, *NLRP7*, and *PADI6)* experience a range of reproductive outcomes, including hydatidiform moles, periods of infertility, reproductive loss, offspring with multi-locus imprinting disorders, and unaffected children [43–46]. Although somewhat anecdotal, it is of interest that one (of five) woman with an *NLRP5* mutation, ascertained following the birth of a child with a multi-locus imprinting disorder, also had a child with an isolated (i.e., apparently non-syndromic) CHD (atrial septal and ventricular defects) [43].

The observed enrichment of genes mapping to GO processes related to ion transmembrane transport, and specifically to calcium ion transport, could also be driven by MEGs. Although a detailed understanding of the genetic regulation of these oscillations is lacking, the known MEG, *NLRP5* (also known as *MATER*), is required for calcium homeostasis. Specifically, oocytes from mouse *Mater* hypomorphs exhibit lower first peak amplitudes and higher frequencies of calcium oscillations (as compared to wild-type oocytes), likely due to a reduction in calcium stores in the endoplasmic reticulum [47].

Our analyses also identified the enrichment of genes related to hypertension. Maternal pregestational hypertension is associated with an increased risk of several birth defects, including, but not limited, to CHDs [48–51]. These associations appear to be independent of medications taken for the treatment of hypertension [48, 49], suggesting either that maternal hypertension, per se, has a negative impact on development (e.g., via an effect on blood flow to the uterus) or that hypertension and birth defects share common risk factors (e.g., genes with pleiotropic effects).

Our analyses also identified enrichment of genes related to proteostasis deficiency and diseases associated with protein misfolding and aggregation (e.g., amyotrophic lateral sclerosis). During pregnancy, the accumulation of misfolded proteins in body fluids and the placenta is associated with preeclampsia, a maternal condition that is also associated with an increased risk of birth defects, including CHDs [50–54].

The results of our study must be viewed in light of both its strengths and limitations. We used a case-control study design, in which we compared the mothers (cases) and fathers (controls) of individuals with CTDs, to identify maternal CTD-related genes. Compared to a case-control design using unrelated, female controls, our design has the advantage of controlling for the potentially confounding effects of the genotype inherited by the child but is subject to bias arising from differences in allele frequencies between males and females. However, sex differences in allele frequencies appear to be uncommon (< 1% of variants) in autosomal genes [55]. In addition, while our analyses assess whether there are gene-level differences between mothers and fathers, they do not indicate which group might carry more (or less)

disease-related alleles. Our observed associations could, therefore, be driven by paternal rather than maternal effects. However, since embryonic development prior to zygotic gene activation is primarily driven by maternal gene products, and the maternal genome has a direct effect on the *in utero* environment, any true associations detected in our study are most likely due to maternal genes. Nonetheless, additional studies (e.g., in model systems) will be required to confirm and establish the mechanisms underlying these observed associations.

Our analyses were based on two large CTD datasets that were ascertained in the United States using similar recruitment, and systematic case confirmation (phenotyping) approaches. Furthermore, our gene-based analyses had a lower multiple-testing burden than SNP-based GWAS. However, our sample sizes were relatively small for our genome-wide approach, and the criterion for achieving statistical significance (corrected-p $\sim 2.5 \times 10^{-6}$) remained quite high. Consequently, associations with maternal CTD-related genes may have been missed in our analyses due to low power. Genes with suggestive evidence of association and genes associated with enriched terms, therefore, appear to be strong targets for further investigations of the maternal genetic contribution to CTDs.

In our analyses, we combined data across different CTD phenotypes, which could have obscured associations if the maternal contribution to individual phenotypes is distinct. However, the predominance of evidence suggests that maternally mediated risk factors tend to be related to a broad spectrum of malformations. For example, maternal hypertension, preeclampsia, diabetes, and obesity are all associated with a spectrum of cardiac and non-cardiac malformations. Hence, for studies of maternal risk factors, the potential for improved power resulting from analyses of similar birth defects (e.g., the various CTD phenotypes) outweighs concerns regarding the potential impact of phenotypic heterogeneity.

This study is the first gene-based GWAS of maternal genotypes and CTDs. We have, however, previously conducted SNP-based, common-variant GWAS and meta-analysis using the same datasets as in the current gene-based analyses [14]. In our SNP-based meta-analysis, we identified several variants with suggestive evidence of association ($p \leq 10^{-5}$); however, none were located in, or within 1kb of the genes with suggestive evidence of association in the current, gene-based analyses. Based on these same two datasets, we have also reported that the risk of CTDs is associated with a maternal genetic risk score for hypertension [10]. However, only one variant included in the genetic risk score falls within a gene that was included in our enrichment analyses (i.e., rs11862778 in *MTHFR*). Hence, these two analyses appear to provide largely independent evidence that genes related to maternal hypertension are associated with the risk of CTDs.

In conclusion, our analyses provide provocative new insights into the potential influence of the maternal genome on embryonic development. While our results are specific to CTDs, both maternal conditions (e.g., hypertension) and MEGs are associated with a range of adverse reproductive outcomes, suggesting that our findings may have much broader implications for the understanding of birth-defect etiology. Further, our findings suggest a link between birth defects and other adverse pregnancy outcomes (e.g., reproductive loss and infertility). Confirmation of such a link would have broad implications for reproductive counseling and planning. Given these initial, compelling findings, additional studies of the relationship between the maternal genome and birth defects are warranted.

## Supporting information

**S1 Fig. Quantile-quantile plot of SKAT-C test gene-based p-values in the CHOP cohort (genomic inflation factor = 1.06).**
(PDF)

**S2 Fig. Quantile-quantile plot of SKAT-C test gene-based p-values in the PCGC cohort (genomic inflation factor = 1.05).**
(PDF)

**S1 File. Table A.** Human homologs of mammalian maternal effect genes identified in reviews by Condic (2016) and Zhang and Smith (2015). **Table B.** A gene-based analysis of the CHOP cohort using the SKAT-C test. The mothers and fathers of patients with CTDs were considered as 'cases' and 'controls,' respectively, for this analysis. **Table C.** A gene-based analysis of the PCGC cohort using the SKAT-C test. The mothers and fathers of patients with CTDs were considered as 'cases' and 'controls,' respectively, for this analysis. **Table D.** A gene-based meta-analysis of SKAT-C test results from the CHOP and PCGC cohort analyses. **Table E.** Gene Ontology (GO) processes from MetaCore enrichment analysis of top genes (meta-analysis $p < 0.01$) from the SKAT-C test, and color-coded GO process clusters identified through REVIGO. **Table F.** Enriched diseases (by biological markers) in the MetaCore enrichment analysis of top genes (meta-analysis $p < 0.01$) from the SKAT-C test.
(XLSX)

## Acknowledgments

Members of the PCGC include (listed alphabetically, by institution): Boston Children's Hospital, Boston, Massachusetts (Jane Newburger and Amy Roberts), Children's Hospital of Los Angeles, Los Angeles, California (Richard Kim), Children's Hospital of Philadelphia (Elizabeth Goldmuntz)\*, Cincinnati Children's Hospital, Cincinnati, Ohio (Eileen C. King), Columbia University Medical School, New York, New York (Wendy Chung), Harvard Medical School, Boston, Massachusetts (Christine Seidman and Jonathan Seidman), Icahn School of Medicine at Mount Sinai, New York, New York (Bruce Gelb), J. David Gladstone Institutes, San Francisco, California (Deepak Srivastava and Daniel Bernstein), New England Research Institutes, Watertown, Massachusetts (Sharon Tennstedt, Kimberly Dandreo, and Julie Miller), Stanford University, Stanford, California (Daniel Bernstein), Steve and Alexandra Cohen Children's Medical Center of New York, New Hyde Park, New York (Angela Romano-Adesman), University of Rochester School of Medicine and Dentistry, Rochester, New York (George Porter), University of Utah, Salt Lake City, Utah (Martin Tristani-Firouzi and H. Joseph Yost), Yale School of Medicine, New Haven, Connecticut (Martina Brueckner and Richard Lifton).

\*Lead PCGC author: goldmuntz@email. chop.edu

## Author Contributions

**Conceptualization:** Anshuman Sewda, Elizabeth Goldmuntz, Deanne Taylor, Laura E. Mitchell.

**Data curation:** Hakon Hakonarson.

**Formal analysis:** Anshuman Sewda, Fadi Musfee.

**Funding acquisition:** Elizabeth Goldmuntz, Hakon Hakonarson, Bernice E. Morrow, Laura E. Mitchell.

**Investigation:** Anshuman Sewda.

**Methodology:** Anshuman Sewda, A. J. Agopian, Laura E. Mitchell.

**Resources:** Hakon Hakonarson.

**Supervision:** Laura E. Mitchell.

**Writing – original draft:** Anshuman Sewda, Laura E. Mitchell.

**Writing – review & editing:** A. J. Agopian, Elizabeth Goldmuntz, Hakon Hakonarson, Bernice E. Morrow, Fadi Musfee, Deanne Taylor.

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
