## [Decision Letter · Decision Letter 0]

7 Apr 2020

PONE-D-20-05538

Gene-based analyses of the maternal genome implicate maternal effect genes as risk factors for conotruncal heart defects

PLOS ONE

Dear Dr. Mitchell,

Thank you for submitting your manuscript to PLOS ONE. After careful consideration, we feel that it has merit but does not fully meet PLOS ONE’s publication criteria as it currently stands. Therefore, we invite you to submit a revised version of the manuscript that addresses the points raised during the review process.

We would appreciate receiving your revised manuscript by May 22 2020 11:59PM. To enhance the reproducibility of your results, we recommend that if applicable you deposit your laboratory protocols in protocols.io, where a protocol can be assigned its own identifier (DOI) such that it can be cited independently in the future. For instructions see: http://journals.plos.org/plosone/s/submission-guidelines#loc-laboratory-protocols

We look forward to receiving your revised manuscript.

Kind regards,

David Scott Winlaw, MBBS MD FRACS

Academic Editor

PLOS ONE

Additional Editor Comments (if provided):

Thank you for submitting this work. The reviewers have raised a number of important concerns and limitations of the study that require addressing and modification of the manuscript, mostly allowing for readers to better understand the context and outcome of the study. Greater clarity in the introduction of why you have have chosen this study methodology, and a broader contextualization of the results for a scientific and medical audience are required before further consideration of publication. There seems to be a gap between the stated aims of the paper, and the validity of the argument in supporting the hypotheses. There is an over-reliance on statistical significance supporting relevance rather than a cohesive argument supporting the role of particular genes in the development of congenital heart disease.

Revision of the manuscript taking into consideration the reviewer comments may strengthen the paper, a point by point rebuttal and outline of changes made will allow further consideration of the paper.

Journal Requirements:

3. One of the noted authors is a group or consortium "Pediatric Cardiac Genomics Consortium". In addition to naming the author group, please list the individual authors and affiliations within this group in the acknowledgments section of your manuscript. Please also indicate clearly a lead author for this group along with a contact email address.

Reviewers' comments:

Reviewer's Responses to Questions

**Comments to the Author**

1. Is the manuscript technically sound, and do the data support the conclusions?

Reviewer #1: Yes

Reviewer #2: Yes

Reviewer #3: No

2. Has the statistical analysis been performed appropriately and rigorously? 

Reviewer #1: Yes

Reviewer #2: Yes

Reviewer #3: No

3. Have the authors made all data underlying the findings in their manuscript fully available?

Reviewer #1: Yes

Reviewer #2: Yes

Reviewer #3: Yes

4. Is the manuscript presented in an intelligible fashion and written in standard English?

Reviewer #1: Yes

Reviewer #2: Yes

Reviewer #3: Yes

5. Review Comments to the Author

Reviewer #1: This study begins to address the interesting and important question of maternal effect gene influence on congenital heart disease in the child. The is the first report of a burden test looking for enrichment in maternal variants over this in the father. The manuscript is well-written and includes a detailed methods section. It also reports some interesting possible associations that could be verified and followed up by subsequent studies. The discussion section is valid on the whole but it is long. Statistical analysis appear to be performed to a high standard. All data presented in this manuscript have been made available and the appropriate ethics appear to have been obtained.

Pathway analysis (Gene Set Enrichment Analysis (GSEA)) has been mentioned (in methods and in the context of meta-analysis) but not it is not clear why GSEA was not performed on individual datasets (as was done for gene-based burden testing). What relevant gene sets were tested?

More importantly, if the aim of this work was to demonstrate a significant enrichment in variants in MEGs in maternal versus paternal genomes, then a MEG list could have tested alone by GSEA, removing the need for multiple comparison correction, and increasing the chance of a significant result.

Did the authors consider filtering those variants included in the burden test based on receiving a high score by pathogenicity predictors such as CADD and polyphen-2, as has been reported in the literature?

In the meta-analysis, the authors report a 2.3-fold enrichment of known MEGs. Could the authors clarify (in the text) if this is significant? It would appear not if the fisher exact figure of p=0.057 pertains to this fold increase. The first sentence of the discussion is not really correct if this is the case.

The term gene-based GWAS is confusing. Use of a more established terms such as rare variant association testing or burden testing could be used and would make it clearer what type of analysis the was being performed. If the authors still prefer the term “gene-based GWAS” then other terms by which it is known (above) could also be provided in brackets. The term Gene Set Enrichment Analysis (GSEA) could also used in places where pathways and gene lists are analysed.

It is somewhat surprising that the placenta is not mentioned in this manuscript. It would seem likely that some MEGs important for CHD include genes relevant to placenta formation, given that disruption of genes in mice that cause placental defects tend to also have heart and vascular defects (Perez-Garcia et al Nature 2018). Comment on the lack enrichment in placental genes could be made in the discussion. Is there a list of genes important for maternal placenta formation that could be used in GSEA?

The author’s rightly point out that true associations are likely maternal rather than paternal associations, however the manuscript does not present those genes enriched for variants in the paternal genomes, which would be useful as a means to judge the number of false associations observed in the analysis.

Reviewer #2: Just a few comments/queries to hopefully improve clarity and transparency:

Page 9 line 181: Replace "an established MEG" with "a previously established MEG"

Page 11 line 218-220 and page 13 lines 261-266: I did not understand why you are including SLAIN2 as one of your "candidate maternal CTD-related genes"? What is it about SLAIN2 (as opposed to the 5 genes on 19q13.2) that puts it into this category?

Page 12 line 239: Replace "provide evidence" with "provide suggestive evidence" or "provide preliminary evidence" or something similar. (As the actual level of evidence seems pretty weak).

Page 13 line 261: Replace "not been implicated" with "not been previously implicated"

Page 16 lines 335-336. It would have been interesting to see results for specific phenotypes (particularly the ones with the largest sample sizes, namely the top 4 phenotypes listed in Table 1). Have you considered doing this? I would not insist on it for this publication, but worth considering in the future...

Page 34 Table 2: Please include the actual p values for the genes with p<10-3 listed (as separate columns, or in brackets after the gene name), as was done in Table 3.

Reviewer #3: Not novel, the authors have recently published a 2019 paper examining the same data set for GWAS/Meta-analyses of CHD in PLOS One (https://doi.org/10.1371/journal.pone.0219926) Not cited in current paper (CHOP trios and PGCG Trios)

While this current manuscript focuses on maternal effect genes, these genes did not reach genome wide significance in the previous paper although the methods highly similar.

It’s worth noting significant portions of the papers overlap with the authors precious publications

In the introduction, the authors state that they have previously conducted SNP based GWAS on this dataset but did not identify any loci with genome wide significance, and thus opted to conducted gene based GWAS as it allowed them to loosen the stringent thresholds for statistical significance while including both common and rare variants.

This coupled with the random urge to look at maternal contributions to CHD (these authors have previously examined maternal contributions in a 2014 PLos One paper and found no significant associations initial 2014 CHOP Trios study, they studied maternal contributions in a 2017 using CHOP trios and LVOTD Trios) seems to be “fishing” especially in light of the 2019 paper

The current paper under review used similar methodology, but looked instead at maternal affect genes, using fathers of the patients as controls. the use of the fathers as controls, completely ignores their contribution to the overarching CHD phenotype making them a less that optimal control especially in light of previous analyses with the proper control samples (2019 paper, 2017 paper and original 2014 paper)

Finally, the authors failed to cite a 2013 CHD GWAS by Cordell et al in Nature Genetics that was very well powered (1995 cases, 5159 controls) that examined 3 major CHD categories together then separately. The authors of this study concluded “Our work, therefore, adds to recent data from studies of CNVs, suggesting that genetic associations with CHD have a considerable degree of phenotypic specificity1”

It appears that the authors of this paper under review have failed to recognize the heterogeneity within the CHD phenotype although in several of the previous papers, they do seem to be aware of it. A 2015 paper by the same group which examined left cardiac malformations via GWAS did identify several associated loci with genome wide significance.

It is not clear why the authors insist on examining the CHD as a single phenotype when in the 2018 cohort description for the PCGC trios, they state that CHD is a “broad spectrum of malformations.”

Further they claim in the manuscript under review the interpretations of their data are hampered by: limited understanding of the mechanisms underlie associations between maternal conditions and birth defects (hello placenta how are you doing? Not well my dear) and lack of detailed annotations specific to the role of maternal genes in offspring development “little is known about mammalian maternally expressed genes” (genes expressed in the egg prior to zygotic gene activation) a google search produced a number of manuscripts on this topic of note is a well cited review by Zhang and Smith 2016, Maternal Control of early embryogenesis in mammals – a review Table 1 lists many genes along with their citations.

6. PLOS authors have the option to publish the peer review history of their article (what does this mean?). If published, this will include your full peer review and any attached files.

Reviewer #1: No

Reviewer #2: No

Reviewer #3: No

---

## [Author Response · Author response to Decision Letter 0]

19 May 2020

Editor Comments

The reviewers have raised a number of important concerns and limitations of the study that require addressing and modification of the manuscript, mostly allowing for readers to better understand… 

See below for our responses to the reviewer comments. We believe that the revisions we have made to the manuscript address the broader concerns expressed by the Editor.

The revised manuscript has been edited to meet PLOS ONE’s style and file naming requirements.

Tables and Figures have been relocated so that they follow the paragraph in which they are first cited. We did not track these changes, as doing so would likely make it more difficult to follow the more substantive changes.

We note that you have indicated that data from this study are available upon request. 

The data used in the analyses described in this manuscript were derived from three studies. These studies have all been registered through dbGap and data from two have been uploaded: 

Pediatric Cardiac Genomics Consortium:

https://www.ncbi.nlm.nih.gov/projects/gap/cgi-bin/study.cgi?study_id=phs001194.v2.p2

CHOP pediatric controls:

https://www.ncbi.nlm.nih.gov/projects/gap/cgi-bin/study.cgi?study_id=phs000490.v1.p1

Data from the third study are being prepared for submission to the related dbGap project.

CHOP CTD trios:

https://www.ncbi.nlm.nih.gov/projects/gap/cgi-bin/study.cgi?study_id=phs000881.v1.p1

This information has been added to the Data Availability section of the manuscript.

One of the noted authors is a group or consortium "Pediatric Cardiac Genomics Consortium". In addition to naming the author group, please list the individual authors and affiliations within this group in the acknowledgments section of your manuscript. Please also indicate clearly a lead author for this group along with a contact email address.

We have moved the individual authors (names and affiliations) from the Pediatric Cardiac Genomics Consortium from Table A of Supplemental Files to the acknowledgements section of the manuscript. A contact author has also been designated and her email has been provided. 

Note: Line references provided below refer to line numbers in the marked-up version of the revised manuscript.

Reviewer #1: 

The discussion section is valid on the whole but it is long. 

We have reduced the level of detail provided in several sections of the Discussion.

Pathway analysis (Gene Set Enrichment Analysis (GSEA)) has been mentioned (in methods and in the context of meta-analysis) but not it is not clear why GSEA was not performed on individual datasets (as was done for gene-based burden testing). 

For all of our analyses, we have emphasized the results from the meta-analysis, because they are (by definition) based on larger numbers and, therefore, provide improved powered to detect associations relative to the individual datasets. However, based on this comment, we realized that our inclusion of the section titled ‘Gene-based GWAS of individual datasets,’ detracted from this emphasis. Consequently, we have removed this section (and modified the related Table 2) from the Results section of the manuscript. We have, however, retained the results of the analyses of the individual studies in the supplemental materials. 

What relevant gene sets were tested?

Gene-set enrichment analyses were performed using MetaCore and included evaluation of MetaCore’s manually-curated gene ontology processes, diseases, pathway maps, and pathway processes (see lines 189-194. The analyses conducted using MetaCore were agnostic (i.e., we did not pre-specify processes of interest).

More importantly, if the aim of this work was to demonstrate a significant enrichment in variants in MEGs in maternal versus paternal genomes, then a MEG list could have tested alone by GSEA, removing the need for multiple comparison correction, and increasing the chance of a significant result.

Our MEG gene-set analysis was not planned a priori, but rather was motivated by the gene-level results in the meta-analysis, where the top gene, GGN, was a proposed MEG. To clarify this point, we have moved the description of the MEG gene-set analysis to a new section in Results, titled ‘Post hoc analyses of maternal effect genes (MEGs)’.

Did the authors consider filtering those variants included in the burden test based on receiving a high score by pathogenicity predictors such as CADD and polyphen-2, as has been reported in the literature?

Our analyses were based on comprehensive coverage of both common and rare variants in each gene. We did not consider or undertake analyses based on CADD or similar scores, as such an approach would dramatically decrease the coverage of each gene.

In the meta-analysis, the authors report a 2.3-fold enrichment of known MEGs. Could the authors clarify (in the text) if this is significant? It would appear not if the fisher exact figure of p=0.057 pertains to this fold increase. The first sentence of the discussion is not really correct if this is the case.

As noted above, this analysis is now clearly identified as being a post hoc analysis motivated by the results of our gene-level analysis. Further, in the Results section, we now state that the observed 2.3-fold enrichments “is of borderline significance (Fisher’s exact p = 0.057)” (lines 322-324). Finally, based on a revision suggested by Reviewer 2, the first sentence of the Discussion section has been modified to read “Our genome-wide, gene-based analyses of common and rare variants provide suggestive evidence…” (modified text in italics).

The term gene-based GWAS is confusing. Use of a more established terms such as rare variant association testing or burden testing could be used and would make it clearer what type of analysis the was being performed. 

Prior to selecting a title for this manuscript, we reviewed related literature and found several publications that used the “gene-based GWAS” terminology. We believe it is important to retain the reference to GWAS, to clarify the scope of our analyses and to include the qualifier, “gene-based”, to distinguish our analyses from SNP-based GWAS. Consequently, we have elected to retain the “gene-based GWAS” terminology. 

If the authors still prefer the term “gene-based GWAS” then other terms by which it is known (above) could also be provided in brackets. The term Gene Set Enrichment Analysis (GSEA) could also used in places where pathways and gene lists are analyzed.

In response to this comment, sections in both Methods and Results that were labeled as ‘Enrichment and gene-set analyses’ have been re-labeled as ‘Gene set enrichment analyses.’

It is somewhat surprising that the placenta is not mentioned in this manuscript. It would seem likely that some MEGs important for CHD include genes relevant to placenta formation, given that disruption of genes in mice that cause placental defects tend to also have heart and vascular defects (Perez-Garcia et al Nature 2018). Comment on the lack enrichment in placental genes could be made in the discussion. Is there a list of genes important for maternal placenta formation that could be used in GSEA?

As noted above, our initial analyses were focused on an agnostic, comprehensive assessment of genes and gene sets. Based on the results of the agnostic gene-level analyses, in which the top gene (GGN) is a suspected MEG, MEGs were assessed in post hoc analyses. As the gene-level analysis did not provide obvious links to the placenta, no post hoc analyses of placental genes were undertaken.

The author’s rightly point out that true associations are likely maternal rather than paternal associations, however the manuscript does not present those genes enriched for variants in the paternal genomes, which would be useful as a means to judge the number of false associations observed in the analysis.

The gene-level analyses identify whether a gene is associated with the outcome but do not provide an estimate of the direction or magnitude of the association. Thus, in this study, the analyses indicate whether (or not) there is a difference between mothers and fathers of cases, but do not indicate which group might carry more (or less) disease-related alleles. We have clarified this point in the Discussion section (text starting on line 409).

Reviewer #2: 

Page 9 line 181: Replace "an established MEG" with "a previously established MEG"

Given the context within which this statement occurs, we believe the addition of “previously” is not required. Specifically, in the same sentence, we define that an established MEG is one that is included in at least one of two published review articles.

Page 11 line 218-220 and page 13 lines 261-266: I did not understand why you are including SLAIN2 as one of your "candidate maternal CTD-related genes"? What is it about SLAIN2 (as opposed to the 5 genes on 19q13.2) that puts it into this category?

In our Statistical Methods section we state that “For genes with at least suggestive evidence of association in the meta-analysis, we consider whether the meta-analysis p-value for the gene was lower than the p-values in the contributing datasets (i.e., the evidence for association was stronger in the combined data than in either of the individual datasets)…” Consequently, SLAIN2 is included as a proposed maternal CTD-related gene because it is the only gene on chromosome 4 with suggestive evidence of association in the meta-analysis, and a meta-analysis p-value that is lower than the p-values obtained in either of the individual studies.

Because the five genes on chromosome 19 (as well as the two genes on chromosome 3) that met these criteria are contiguous, it is likely that the observed associations are not independent, but rather reflect linkage disequilibrium between variants in different genes. Consequently, for each of these two regions, we elected the gene for which we believe there is the strongest evidence that it might act through the maternal genotype. We have revised the Method section (text starting on line 177) to clarify our rationale for the selection of our candidate maternal CTD-related genes in these two regions.

Page 12 line 239: Replace "provide evidence" with "provide suggestive evidence" or "provide preliminary evidence" or something similar. (As the actual level of evidence seems pretty weak).

We have revised this text by adding the word “suggestive”.

Page 13 line 261: Replace "not been implicated" with "not been previously implicated"

We have made this suggested revision.

Page 16 lines 335-336. It would have been interesting to see results for specific phenotypes (particularly the ones with the largest sample sizes, namely the top 4 phenotypes listed in Table 1). Have you considered doing this? I would not insist on it for this publication, but worth considering in the future...

We agree that analyses of individual CTD phenotypes would be of interest. However, particularly when assessing maternal genotypes, we believe that there is strong rationale for analyzing CTDs as a group (see Discussion, paragraph starting on line 447). For these reasons, and concerns about small sample sizes, we have not conducted analyses of the individual phenotypes.

Page 34 Table 2: Please include the actual p values for the genes with p<10-3 listed (as separate columns, or in brackets after the gene name), as was done in Table 3.

In response to a comment from Review 1, we have opted to focus on the findings from the meta-analysis. Consequently, Table 2 has been modified and no longer includes the list of suggestive genes. Exact p-values for all genes assessed in the individual studies have, however, been retained in the supplemental materials.

Reviewer #3: 

Not novel, the authors have recently published a 2019 paper examining the same data set for GWAS/Meta-analyses of CHD in PLOS One 

The referenced paper is based on the same datasets, but describes analyses of the association between CTDs and the inherited genotype. Consequently, the analyses described in that paper are not directly relevant to the analyses presented in this paper. We have, however, provided a reference for our prior SNP-level GWAS of the maternal genotype and CTDs (Agopian et al 2017).

While this current manuscript focuses on maternal effect genes, these genes did not reach genome wide significance in the previous paper although the methods highly similar.

True, but we would not expect that maternal genes associated with CTDs would be the same genes that are associated with CTDs via the inherited genotype. 

It’s worth noting significant portions of the papers overlap with the authors precious publications

The analyses described in this paper are based on the same datasets that have been used in prior publications. Thus, there is overlap in the description of the study subjects and data collection and processing procedures. As we have not previously published on gene-level analyses of the maternal genotype, the results presented in this manuscript do not overlap with any of our prior work.

In the introduction, the authors state that they have previously conducted SNP based GWAS on this dataset but did not identify any loci with genome wide significance, and thus opted to conducted gene based GWAS as it allowed them to loosen the stringent thresholds for statistical significance while including both common and rare variants.This coupled with the random urge to look at maternal contributions to CHD (these authors have previously examined maternal contributions in a 2014 PLos One paper and found no significant associations initial 2014 CHOP Trios study, they studied maternal contributions in a 2017 using CHOP trios and LVOTD Trios) seems to be “fishing” especially in light of the 2019 paper.

In the introduction, we state that GWAS “provide a comprehensive, agnostic approach”. While this approach may be viewed as “fishing”, GWAS have had a major, positive impact on our understanding of the genetic contribution to complex traits. 

The suggestion that our evaluation of the maternal contribution to CTD risk represents a “random urge” is incorrect. Evaluation of the maternal genotype is a topic of some interest within the birth defects research community and has been a major research focus for our group for over two decades. Our work in this domain has been and continues to be funded by the NIH, and we have multiple publications describing both our methodologic and applied work on this topic.

The current paper under review used similar methodology, but looked instead at maternal affect genes, using fathers of the patients as controls. the use of the fathers as controls, completely ignores their contribution to the overarching CHD phenotype making them a less that optimal control especially in light of previous analyses with the proper control samples (2019 paper, 2017 paper and original 2014 paper)

We assume that the reviewer is referring to our prior published studies. As noted above, some of these publications (e.g., Sewda et al. 2019) are based on studies of the inherited genotype and are not relevant to this submission. Our prior SNP-based studies of the maternal genotype (2014 and 2017) used a trio-based approach in which the maternal genotype is assessed by comparison of reciprocal mating types (e.g., mother AA x father aa versus mother aa versus father AA) – so these comparisons were also based on data from mothers and fathers. 

We have expanded our Discussion of the strengths and limitations of our analytic approach in response to the last comment provided by Reviewer 1 (see response above and paragraph starting on line 402). In this section, we indicate that the comparison of mothers to fathers, in a case-control framework, appropriately controls for the correlation between the parental and inherited genotypes. Specifically, because the parent-child genetic correlation is the same for mothers and for fathers, genes that are associated with CTDs solely through the inherited genotype would not be identified using this design.

Finally, the authors failed to cite a 2013 CHD GWAS by Cordell et al in Nature Genetics that was very well powered (1995 cases, 5159 controls) that examined 3 major CHD categories together then separately. The authors of this study concluded “Our work, therefore, adds to recent data from studies of CNVs, suggesting that genetic associations with CHD have a considerable degree of phenotypic specificity1” 

The Cordell paper provides an evaluation of the inherited genotype and, therefore, is not directly relevant to this study.

It appears that the authors of this paper under review have failed to recognize the heterogeneity within the CHD phenotype although in several of the previous papers, they do seem to be aware of it. A 2015 paper by the same group which examined left cardiac malformations via GWAS did identify several associated loci with genome wide significance.

It is not clear why the authors insist on examining the CHD as a single phenotype when in the 2018 cohort description for the PCGC trios, they state that CHD is a “broad spectrum of malformations.”

See our response to the second to last comment from Reviewer 2.

Further they claim in the manuscript under review the interpretations of their data are hampered by: limited understanding of the mechanisms underlie associations between maternal conditions and birth defects (hello placenta how are you doing? Not well my dear) and lack of detailed annotations specific to the role of maternal genes in offspring development “little is known about mammalian maternally expressed genes” (genes expressed in the egg prior to zygotic gene activation) a google search produced a number of manuscripts on this topic of note is a well cited review by Zhang and Smith 2016, Maternal Control of early embryogenesis in mammals – a review Table 1 lists many genes along with their citations.

We have removed the referenced text from the revised manuscript and have moved the remainder of the text from that paragraph to an earlier section of the discussion (lines 365-375).

While revising your submission, please upload your figure files to the Preflight Analysis and Conversion Engine (PACE) digital diagnostic tool, https://pacev2.apexcovantage.com/. PACE helps ensure that figures meet PLOS requirements. To use PACE, you must first register as a user. Registration is free. Then, login and navigate to the UPLOAD tab, where you will find detailed instructions on how to use the tool. If you encounter any issues or have any questions when using PACE, please email us at figures@plos.org. Please note that Supporting Information files do not need this step

Figure 1 has been evaluated for PLOS requirements using PACE. The remaining figures appear in the Supporting Information files.

---

## [Editor Report · Decision Letter 1]

26 May 2020

Gene-based analyses of the maternal genome implicate maternal effect genes as risk factors for conotruncal heart defects

PONE-D-20-05538R1

Dear Dr. Mitchell,

We are pleased to inform you that your manuscript has been judged scientifically suitable for publication and will be formally accepted for publication once it complies with all outstanding technical requirements.

With kind regards,

David Scott Winlaw, MBBS MD FRACS

Academic Editor

PLOS ONE

Additional Editor Comments (optional):

Thank you for revising your manuscript, this will make an excellent contribution.
---

## [Editor Report · Acceptance letter]

29 May 2020

PONE-D-20-05538R1 

Gene-based analyses of the maternal genome implicate maternal effect genes as risk factors for conotruncal heart defects 

Dear Dr. Mitchell:

I am pleased to inform you that your manuscript has been deemed suitable for publication in PLOS ONE. Congratulations! Your manuscript is now with our production department. 

With kind regards,

on behalf of

Professor David Scott Winlaw 

Academic Editor

PLOS ONE